# Reassessing the Abundance of miRNAs in the Human Pancreas and Rodent Cell Lines and Its Implication

**DOI:** 10.3390/ncrna9020020

**Published:** 2023-03-17

**Authors:** Guihua Sun, Meirigeng Qi, Alexis S. Kim, Elizabeth M. Lizhar, Olivia W. Sun, Ismail H. Al-Abdullah, Arthur D. Riggs

**Affiliations:** 1Department of Diabetes Complications & Metabolism, Arthur Riggs Diabetes & Metabolism Research Institute, City of Hope, Duarte, CA 91010, USA; 2Department of Neurodegenerative Diseases, Beckman Research Institute, City of Hope, Duarte, CA 91010, USA; 3Department of Translational Research & Cellular Therapeutics, Arthur Riggs Diabetes & Metabolism Research Institute, City of Hope, Duarte, CA 91010, USA; 4Department of Diabetes & Cancer Metabolism, Arthur Riggs Diabetes & Metabolism Research Institute, City of Hope, Duarte, CA 91010, USA

**Keywords:** miRNA, miRNA-375, miRNA-7-5p, miRNA-148a-3p, islet, acinus

## Abstract

miRNAs are critical for pancreas development and function. However, we found that there are discrepancies regarding pancreatic miRNA abundance in published datasets. To obtain a more relevant profile that is closer to the true profile, we profiled small RNAs from human islets cells, acini, and four rodent pancreatic cell lines routinely used in diabetes and pancreatic research using a bias reduction protocol for small RNA sequencing. In contrast to the previous notion that miR-375-3p is the most abundant pancreatic miRNA, we found that miR-148a-3p and miR-7-5p were also abundant in islets. In silico studies using predicted and validated targets of these three miRNAs revealed that they may work cooperatively in endocrine and exocrine cells. Our results also suggest, compared to the most-studied miR-375, that both miR-148a-3p and miR-7-5p may play more critical roles in the human pancreas. Moreover, according to in silico-predicted targets, we found that miR-375-3p had a much broader target spectrum by targeting the coding sequence and the 5′ untranslated region, rather than the conventional 3′ untranslated region, suggesting additional unexplored roles of miR-375-3p beyond the pancreas. Our study provides a valuable new resource for studying miRNAs in pancreata.

## 1. Introduction

The pancreas is an essential metabolic organ composed of, among other cell types, endocrine and exocrine cells that undergo unique stages during differentiation. This process is controlled by the expression of specific transcription factors. Multipotent progenitor cells in the pancreatic bud first differentiate into tip progenitors which then develop into acinar cells and trunk progenitor cells. The latter further develop into ductal cells, another major type of exocrine cells, and endocrine progenitor cells. Next, endocrine progenitor cells differentiate into islets of Langerhans containing glucagon-producing alpha, insulin-producing beta, somatostatin-producing delta, ghrelin-producing epsilon, and pancreatic polypeptide-producing PP cells, which produce essential hormones to control glucose homeostasis. Endocrine cell differentiation is controlled by Neurogenin 3 (*NEUROG3* or *NGN3*) and other islet cell-specific factors [1,2]. While many transcription factors essential to the process of pancreatic development and function have been identified (listed in Appendix A) [2,3,4], pancreatic development and function can also be modulated by other regulatory molecules such as microRNAs (miRNAs).

Long and short non-coding RNAs, in concert with coding RNAs, function to orchestrate gene regulation [5,6]. miRNAs are small non-coding RNAs with biological activities [7]. miRNAs are known to modulate important processes in the pancreas both in models of good health and disease [8,9,10]. However, an analysis of published high-throughput sequencing datasets of small RNAs from the pancreas has revealed inconsistent results in the population of pancreatic miRNAs and their expression levels, especially miRNAs that are highly expressed in pancreata.

In this study (events were depicted in Figure 1A), using a previously developed bias-reduction small RNA deep sequencing (smRNAseq) protocol, we profiled small RNA from eight pairs of human acinar and islet cells and identified miR-375-3p (hereinafter referred to as miR-375), miR-148a-3p, and miR-7-5p as the most highly abundant miRNAs in human pancreatic cells, in contrast to the generally accepted idea that miR-375 was the most abundant pancreatic miRNA that has been most studied. Because the four rodent pancreatic cell lines, including mouse alpha-TC1, beta-TC-6, MIN6, and rat INS-1, have been routinely used in pancreatic research and diabetes studies, we also profiled small RNAs in these cells. Due to the difficulties in elucidating the potential roles of these three miRNAs in human pancreas cell development and differentiation in vivo, in the current research, we performed in silico studies of miRNA-target interactions between these three miRNAs and their predicted and validated targets in pancreatic genes. According to their predicted targets, we found that both miR-148a-3p and miR-7-5p had a broader target spectrum than miR-375-3p for target sites located in the traditional 3′ untranslated region (3′UTR) that are favored by most miRNAs. However, when taking the target sites in the coding sequence (CDS) or the 5′ untranslated region (5′UTR) into consideration, miR-375-3p has a much broader target spectrum than miR-148a-3p or miR-7-5p. Most predicted miR-375 target sites are in CDS or 5′UTR. This result suggests that miR-375 may have additional unexplored roles in the pancreas and beyond it. According to their validated targets in highly expressed pancreatic genes and essential pancreatic genes (major transcription factors and key products), all three miRNAs play a critical role in the pancreas. Pathway analysis using the above-validated targets showed that miR-7-5p plays a more significant role in insulin pathways than miR-148a-3p, and miR-375-3p plays a less important role among the three miRNAs. We hope the results from this study will provide a metric for future in vivo studies of the miRNA regulation of human pancreatic development, differentiation, and function.

## 2. Results

### 2.1. The Results of Published miRNA Profiles Indicate Discrepancies for Both miRNAs Expressed in Pancreases and Their Abundance in These Datasets

About two decades ago, a large number of miRNAs were identified and cloned. Since then, numerous miRNA profiling datasets in different species and organs have been generated by different techniques and documented. Among them, the small RNA deep sequencing results are highly appreciated, as the sequencing approach can simultaneously profile almost all species of small RNAs as well as different isoforms of a given small RNA. However, there are limited human pancreas small RNA profiling data generated by smRNAseq and other techniques in the GEO database, probably due to the difficulty of accessing fresh human tissues in general. We reanalyzed one dataset containing smRNAseq results from both human whole islets and sorted alpha and beta cells (GSE52314 by Kameswaran et al.) [11]. The reanalysis employed miRge [12,13] and the analyzed results were compared to the published results. Due to the low sequencing depth for the three islet datasets (SRR1028929, SRR1028930, and SRR1028931) in GSE52314, they were pooled and treated as the miRNA profile of islets (hereafter referred to as miRNAs of islets). Because refined profiling methodologies (including both the sequencing device and the library construction protocol for smRNAseq) were used for sorted alpha and beta cell samples, datasets from alpha (SRR1028924) and beta (SRR1028925) cells have much higher sequencing depths than all three islet samples. Because alpha and beta cells account for the majority of human islet cells, we treated the average of the datasets of alpha and beta cells as the estimated miRNA profile of islets (hereafter referred to as estimated miRNAs of islets). Although our reanalysis of islet smRNAseq data agreed with published data in general (Figure 1B, “miRge” versus “Ori ref” in “Islets”, “Alpha”, “Beta”), indicating that the approach of our smRNAseq data analysis performs similarly to the approach used in the data analysis in original publications, there are differences between the miRNAs of islets (Figure 1B, “Islets”) and the estimated miRNAs of islets (Figure 1B, “Alpha + Beta”). A profile of miRNAs of islets revealed that let-7 family members accounted for over 50% of the total miRNAs and miR-375 was the second-most abundant miRNA, representing 13% of the total miRNAs in the pancreas. In contrast, in the estimated miRNAs of islets, the most abundant miRNA was miR-375, accounting for ~38% of total miRNAs, while miR-7-5p was the second-most abundant miRNA, representing 7% of total miRNAs in the islets. The percentage drop in let-7 family members in the estimated miRNAs of islets was accompanied by a percentage increase in several miRNAs, including miR-127-3p, miR-191-5p, miR-99b-5p, miR-26, miR-27, miR-125a-5p, and miR-148a-3p. While there is a possibility that the sorting of alpha and beta cells from islets may have an effect on their profiling results, deeper sequencing methodology used for sequencing alpha and beta cells likely accounts for these differences, and the estimated miRNAs of islets may be closer to the true miRNA profiles of islets. There are also differences between the published data and our reanalyzed data, mainly in the abundance of let-7 family members, as indicated by the ratio of miRNAs in alpha cells and beta cells (Figure 1B). These variations may have arisen from the use of different mapping algorithms with different mismatch requirements, which could generate different total read counts for an miRNA with multiple family members. To rule out potential problems that could have been caused by analysis tools for small RNA profiling, we reanalyzed another dataset that contained whole islets and sorted the beta cells (GSE47720 by van de Bunt et al.) [14]. Then, we compared our analyzed data with the curated data in the miRmine database that contains analyzed data for all six runs in GSE47720 that have been generated using different data analyzing tools [15]. We first compared runs with high sequence reads, specifically SRR873381 for islets and SRR873401 for beta cells, which are likely more reliable due to their higher sequencing depth. Both results analyzed by miRge and miRmine agreed well with each other, further indicating that our analyzed result is not biased by the approach we have used to analyze smRNAseq data. Although miR-375 is still shown to be the most abundant miRNA in the pancreas and covered more than 40% of total miRNA populations in the pancreas, the profiling data showed several differences compared to GSE52314. Notably, miR-7-5p, a miRNA that covered ~7% of the total miRNA reads from the pancreas and was the second-most abundant miRNA in the estimated miRNAs of islets in GSE52314 (Figure 1A, “Alpha + Beta”), dropped to ~0.03% in the dataset of GSE47720, representing an over 200-fold reduction in abundance (Figure 1B, “Islets” and “Beta”). The low expression level of miR-7-5p in the dataset of GSE47720 also conflicts with the published qPCR result showing that miR-7-5p was an islet-enriched miRNA [16] and an islet-specific microRNA during human pancreatic development [17]. Next, we compared the average of datasets in GSE47720 from all three runs (SRR873381, SRR871609, and SRR871652) for islets to the average of datasets from all three runs (SRR873401, SRR873410, and SRR871601) for beta cells, regardless of the sequencing depth, using the analyzed data by miRmine (Figure 1C, “Islets-all” and “Beta-all”). In the data averaged from our combined runs, miR-7-5p accounted for 4 to 5% of total miRNAs in the islets, which represents a greater than 100-fold increase in abundance than in the deeper runs alone, again highlighting the possible impact of different sequencing methodologies on the outcome of miRNA abundance. Furthermore, pooling data from runs using different sequencing protocols may complicate the profiling outcomes, as in the case of miR-143-3p, which was found to be the second-most abundant islet miRNA (16.9% of the total) in the pooled data. Published miRNA profile data acquired via arrays also showed that, while miR-375 was the most abundant miRNA in alpha cells and beta cells, miR-7-5p was not among the 10 most plentiful types [18], suggesting that result heterogeneity can also be found in analyses where miRNA microarray techniques were employed. We also looked at miRNA profiling data in a human tissue miRNA database (miRNATissueAtlas2) and found that the expression of miR-375 is ranked 5th at 16 thousand reads per million mapped reads (RPM), behind the 26.4 thousand RPM of let-7b-5p, 35.4 thousand RPM of miR-26a-5p, 37.3 thousand RPM of let-7a-5p, and 98 thousand RPM of miR-143-3p in the human pancreas, despite being considered pancreas-specific and the most highly expressed pancreatic miRNA [19]. Therefore, it is possible that other small RNA profiling datasets for the pancreas also contain inconsistent small RNA profiling data and this kind of inconsistency in data may also exist in datasets of other human tissues. Despite the inconsistent miRNA profiling data from GSE47720 and GSE52314, the two datasets were extensively used for several miRNA functional studies [8,11,14,20,21,22]. Therefore, to provide a better metric for miRNA abundance in the pancreas for future studies on miRNA function in the human pancreas, a less biased miRNA profile from high-quality tissues closer to the true profile and more relevant to related studies is needed.

**Figure 1 ncrna-09-00020-f001:**
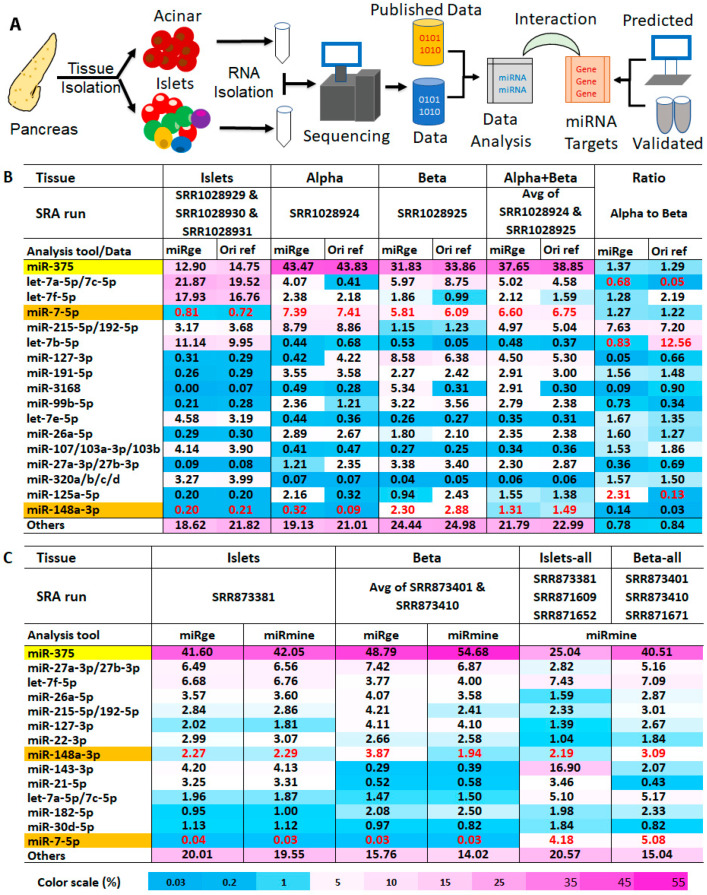
Bias in published pancreatic miRNA profiling data. (**A**) Flowchart of events in our experiment; (**B**) Top expressed miRNAs (by percentage of total miRNA) in islets, alpha, and beta cells in the dataset of GSE47720/PRJNA193453 as described in the original publication and reanalyzed by miRge; (**C**) Top expressed miRNAs (by percentage of total miRNA) in islets and beta cells in the dataset of GSE52314/PRJNA227380 that were reanalyzed by miRge or miRmine. “Islets” and “Beta” represent data from runs with high sequencing depth and “Islets-all” and “Beta-all” represent average RPM from all three runs for islets and beta cells regardless of sequencing depth. All data in both tables A and B are in percentage, 1% represents RPM at 10,000. To permit comparison by different analysis tools and data sources, the RPM of an miRNA was defined as the reads of an miRNA per one million miRNA reads. This was not normalized to all reads mapped to the genome in a deep sequencing run. Abbreviations: GEO = Gene Expression Omnibus, GSE = GEO Series; SRA = Sequence Read Archive; SRR = SRA run number; ref = reference; miRge and miRmine are two computing tools that were used to analyze small RNA deep sequencing data.

### 2.2. Less Biased Small RNA Profiles in Human Acinar and Islet Cells

Variations in pancreatic miRNA profiling data can arise from numerous causes, such as the different extents of RNA degradation in different samples related to the freshness, cause of death, and heterogenicity of the samples, as well as the methods used to acquire and process samples and the robustness of the sequencing technique [23]. It is very challenging to get high-quality fresh human samples due to the nature of human samples in that they need to be from brain-dead cadaveric donors that have gone through many regulation and processing steps, including hospitalization, transportation, and isolation prior to arriving at research laboratories. Among all the confounding factors, smRNAseq methodologies were found to be the major cause of result bias in small RNA profiling. Possible reasons for smRNAseq data heterogeneity include ligase bias towards certain sequences, biased PCR amplification of ligated products, unknown effects of small RNA modification on ligation or reverse transcription, unexpected small RNA sequence interactions with cloning adaptor sequences, and the use of different algorithms for analyzing smRNAseq data [23,24,25]. Even bearing these aspects in mind, a 10 to 100-fold variation in miRNA levels is not acceptable as these variations could confound conclusions from studies using datasets from such miRNA profiling and mislead the designs of future experiments depending on results from such profiling. To this end, we sequenced small RNAs in acinar cells and corresponding islets from human pancreata of eight organ donors using a bias reduction protocol for smRNAseq that we previously developed (Table 1) [25,26]. Most importantly, we have taken advantage of the unique opportunity of having an islet transplantation program in our institute to obtain high-quality human samples as fresh as possible and at relatively large numbers for the current study (Appendix A).

Historically, miR-375 was thought to be the most abundant pancreatic miRNA, which is supported by many of the published pancreatic miRNA profiles, and is the most studied pancreatic miRNA, especially using mouse models. However, the analysis of miRNA abundance in our smRNAseq dataset, as reads per million mapped reads (RPM), revealed that miR-148a-3p, miR-375, and miR-7-5p probably were the most abundant pancreatic miRNAs (Figure 2A). We found that miR-148a-3p was the most abundant miRNA in human acinar cells and miR-148a-3p, miR-375, and miR-7-5p were the most abundant miRNAs in human islets. miR-26a/b-5p and miR-27a/b-3p were equally well expressed in acinar cells and islets. In contrast, miR-148a-3p, miR-21, and miR-217 were significantly lower in islets compared to acinar cells. The abundant miRNAs, miR-148a-3p, miR-375, and miR-7-5p also showed great variation in their expression levels between acinar cells and islets. miR-148a-3p was reduced from 48.25% in acinar cells to about 15.92% in islets, while miR-375 was increased from 4.75% in acinar cells to 16.91% in islets, and miR-7-5p was increased from <0.7% in acinar cells to 20.18% in islets (Figure 2B,C, Appendix A).

We validated the smRNAseq results using small RNA qRT-PCR (Figure 2D). The small RNA qRT-PCR results were normalized to the miR-26a-5p expression in each sample and were compared to normalized results in the smRNAseq data (ratio of RPM of a miRNA to RPM of miR-26a-5p). miR-26a-5p was chosen for normalization because it has a similar expression in acini and islets and is highly expressed in both tissues (Figure 2A). The qRT-PCR data agreed well with smRNAseq data for the relative expression level of each miRNA in acinar cells versus islets (Figure 2D,E). However, there are instances where the smRNAseq data and qRT-PCR data do not agree very well. For example, qRT-PCR showed that both miR-375 and miR-21-5p exhibited a relatively higher expression level compared to smRNAseq. It has been a known issue that small RNA qRT-PCR data does not always agree with deep sequencing data. While qRT-PCR data are usually normalized to an miRNA or several miRNAs, deep sequencing data have the power to be normalized to all miRNA reads or all small RNA reads in a deep sequencing run. In this regard, deep sequencing data will have the advantage of representing the relative abundance of an miRNA. The other reason might be due to the nature of smRNAseq technology which can detect multiple isoforms of an miRNA. While smRNAseq with high sequencing depth may survey many isoforms of an miRNA, the detection of a certain high expression level sequence may be saturated. Conversely, due to the short length of mature miRNA sequences, qRT-PCR primarily detects one specific isoform with higher detecting ranges.

Since we have a relatively large small RNA profiling dataset, we also attempted to identify novel pancreatic miRNAs in these samples, especially those that are highly expressed. Several candidate novel pancreatic miRNAs were predicted using miRge2 (sequence library matches sequences in miRBase Release 22.1, May 2018) [12]. Some of these predicted miRNAs were specifically expressed in acini or islets (Appendix A).

We characterized four predicted pancreatic miRNAs that had high read counts (Appendix A). We designated these miRNAs miR-P1 to P4. miR-P1 was found at a 3 to 1 ratio in acini versus islets, and miR-P4 was found at a 1 to 1 ratio in acini versus islets. Both miR-P2 and miR-P3 were found only in islets. We used their mature sequences to query miRNAs in the latest version of miRBase (Release 22.1, October 2018, June 2021) and noted that miR-P1 matched with miR-802-3p, miR-P2 matched with hsa-mir-7-3-5p, miR-P3 matched with hsa-mir-153-1-5p, and miR-P4 matched with hsa-mir-452-3p. These miRNAs were likely deemed novel by miRge due to their precursor sequences. Alternatively, these miRNAs may have been included in the latest version of miRBase that was released after the release of miRge2 and updated thereafter [12,27]. Nonetheless, our data indicate that the most highly expressed miRNAs in the pancreas have probably been identified already. Additionally, miR-P2 results support our finding that miR-7-5p was present in greater amounts in islets compared to acini.

Combining smRNAseq data from acini and islets together and classifying them by miRNA families, we found that miR-148a-3p was the most abundant miRNA in the human pancreas, followed, from the most to the least, by miR-375, miR-7-5p, miR-26-5p, let-7-5p, miR-21-5p, miR-30-5p, miR-200-3p, miR-27-3p, miR-143-3p, miR-217, miR-99-5p, miR-215-5p/192-5p, and miR-101-5p. Together, these miRNAs formed the top 14 abundant miRNAs/miRNA families in the pancreas (Figure 3A). Differentially expressed gene analysis using DESeq2 showed that there were many other miRNAs differentially expressed between acini and islets (Figure 3B). Cluster and heatmap analyses showed that miR-148a-3p, miR-375, and miR-7-5p were clustered together as the miRNAs with the highest expression, followed by the second-most highly expressed miRNA cluster, which included miR-30a-5p, miR-217, miR-127-3p, miR-221-3p, miR-148a-5p, and miR-216b-5p (Figure 3C). Interestingly, miR-7-5p and miR-217 had the greatest change in expression level between acinar cells and islets in both qRT-PCR and smRNAseq, consistent with miRNA microarray expression data in rat acinar cells and islets [16]. This result suggests a potential role for these miRNAs in the separation of tip progenitors from trunk progenitors.

### 2.3. Comparing Small RNA Profiles in Human Acini and Islets with Small RNA Profiling Data from Rodent Acini and Islets

Rodents have been widely used in pancreas and endocrine research despite differences in tissue cytoarchitecture, endocrine cell proportions, and signaling between human and murine cells and organs [28]. We compared our human miRNA profiling data with a set of published murine miRNA profiles summarized in a comprehensive review of miRNAs and their functions in the pancreas [8]. Specially, we compared our profiling results of human acinar cells and islets with mouse acinar miRNA profiling data in GSE81260 [29] and mouse islet miRNA profiling data (no raw dataset in GEO) [30].

As observed in our human acinar cell miRNA profiling, mouse miR-148a-3p (mmu-miR-148a-3p) was the most abundant miRNA in murine acini. However, while the hsa-miR-26 family is the second-most abundant miRNA in human acinar cells, mmu-miR-375 is the second-most abundant miRNA in murine acini (Appendix A). As observed in our human islet miRNA profiling, miR-148a-3p, miR-375, and miR-7-5p are also the three most abundant miRNAs in murine islets (Appendix A).

Due to their higher rate of proliferation and insulin production, rodent cell lines were more routinely used in pancreas and diabetes research than human cell lines. Therefore, we also profiled small RNAs in four widely used rodent cell lines, murine pancreatic alpha cell lines alpha-TC1, murine beta cell line beta-TC-6 and MIN6, and rat beta-like INS1 cell line.

Our profiling results showed that the three mouse cell lines, alpha-TC1, beta-TC-6, and MIN6, had similar miRNA profiles. However, there are major differences between our miRNA profiling data and the published MIN6 cell smRNAseq data in GEO (GSE44262), with the exception of miR-375 being shown to be the top expressed miRNA in our profiling results for all three murine cell lines and the results for MIN6 cells from GSE44262 [20] (Appendix A). In our profiling data, mmu-miR-7-5p and mmu-miR-26a-5p were the second- and third-most abundant miRNAs in all three cell lines, whereas data for MIN6 cells from GSE44262 showed that mmu-miR-7-5p was expressed at very low levels compared to mmu-miR-375. Instead, mmu-miR-182-5p and mmu-miR-27b-3p were ranked the second and third most abundant miRNAs in data for MIN6 cells from GSE44262 (Appendix A). Notably, mmu-miR-148a-3p was expressed at low levels in these murine insulinoma cell lines, consistent with its role as a tumor suppressor gene that is downregulated in cancers [31,32,33].

In the rat beta-like INS1 cell line, rat miR-375 (rno-miR-375), rno-miR-7-5p, rno-miR-26a-5p, and rno-miR-148-3p were the four most abundant miRNAs similar to their abundance in human pancreas samples (Appendix A and Figure 3A). INS-1 is also an insulinoma cell line. However, unlike mmu-miR-148a-3p expression in the three murine cell lines, rno-miR-148-3p was expressed at a relatively high level (Appendix A). Published array data also showed that rno-miR-217 is abundant in rat acinar cells versus islets and agrees with our data on human acinar cells and islets [16].

In summary, our human pancreatic miRNA profiling data is consistent with published profiling data from rodent cells regarding dominant miRNAs in both acinar cells and islets. The presence of mmu-miR-7-5p in our MIN6 profiling data, an miRNA that was discriminately profiled by different smRNAseq methodologies, as indicated in our reanalysis of published data, indicated that our rodent cell line profiling may also be closer to their true expression level. However, we acknowledge that we only performed one replicate of sequencing for each murine cell line and the two donated cell lines (MIN-6 and INS-1) by other laboratories were not characterized for their identification.

### 2.4. Potential Roles of miR-375, miR-7-5p, and miR-148a-3p in the Human Pancreas According to Their Predicted Targets

Acinar cells and islets are derived from tip and trunk progenitor cells, respectively [1,2]. Conceivably, differentially expressed miRNAs between acini and islets, such as miR-148a-3p, miR-7-5p, and miR-217, may play a role in cell fate decisions during this differentiation process [9]. In general, miRNAs function by regulating specific target molecules, such as transcription factors, by targeting their 3′UTR [34]. A number of transcription factors that participate in pancreatic development and differentiation were identified and described [1,2,3,4]. Based on this data, we generated a list of pancreatic transcription factors and functional genes and designated them as pancreas-essential genes (Appendix A).

Our initial analysis included targets predicted by the TargetScan algorithm (version 7.2 and 8.0), which used the seed sequence of an miRNA or a family of miRNAs (nucleotides #2 to 8 from 5′ end) to locate conserved complementary sequences in the 3′UTR regions of genes [35]. Targeting conserved sequences that cross several species is an indication of a bona fide target of an miRNA. Unexpectedly, miR-375, which has been widely studied and is believed to be critical for and specific to the pancreas, was found to target only a few of the designated pancreas-essential genes [36,37]. TargetScan predicted more targets for miR-148a-3p and miR-7-5p than for miR-375 with several among the pancreas-essential genes (Figure 4A). Therefore, we looked at miRNA targets in miRDB which include both predicted and validated targets [38]. miRDB (version 2020) gave similar results in that the number of targets of miR-148a-3p and miR-7-5p were greater than the number of targets of miR-375 (Figure 4B). Next, upon employing miRWalk, which covers all seed matched sites in CDS, 5′UTR, and 3′UTR [39], we found that the target sites of miR-148a-3p and miR-7-5p predicted by miRWalk (new version 2022) compared to the target sites predicted by TargetScan (version 8.0) increased by about 10-fold (from about 1000 to about 10,000), but the number of miR-375 target sites increased more than 70-fold (from about 500 to about 36,000). The increase in miR-375 targets was presumably due to most new targeting genes having miR-375 target sites located in their CDS or 5′ UTR regions. When seed sequences in CDS, 5′UTR, and 3′UTR were counted, more pancreas-essential genes were noted to be potential targets of miR-375 compared to miR-148a-3p and miR-7-5p (Figure 4C). This result suggests that miR-375 may employ a different targeting mechanism than the other two miRNAs. In contrast to the other two miRNAs, which are also highly expressed in other organs or tissues, miR-375 is only highly expressed in the pancreas, suggesting that miR-375 may be the most important pancreatic miRNA even though it is not the sole dominant miRNA in the pancreas. It is also conceivable that circulating miRNA-375, usually found in serum and exosomes, may behave like insulin, is promiscuous, and may signal beyond its function in the pancreas [40,41,42]. The broad target spectrum of miR-375 may be the result of evolution selection between miRNA-375 and its broad targets in the pancreas and beyond.

The Venn Diagram analysis did not yield a single essential gene targeted by all three miRNAs (Figure 4). To focus on genes with conserved miRNA target sites that have a better chance to have biological functions, TargetScan was employed to assess the predicted genes of miR-375, miR-148a-3p, and miR-7-5p that are also pancreatic cell type-specific genes identified by single cell RNA sequencing in each pancreatic cell group [3,4]. Coinciding with its greater abundance in the pancreas, TargetScan (version 7.2) predicted more pancreas-essential gene targets of miR-148a-3p than miR-375 or miR-7-5p in the nine major pancreas cell types (Appendix A). Here, we also discovered several essential pancreatic genes targeted by more than one miRNA. For example, miR-375 and miR-7-5p were predicted to target *GATA-6* in acinar and ductal cells and *PAX6* in alpha and PP cells. miR-375 and miR-148a-3p were predicted to target *klf5* in acinar and ductal cells and *ELAVL4* in beta and delta cells. miR-148a-3p and miR-7-5p may target *EGFR* and *ZNF704* in acinar and ductal cells and *MAP1B* in beta and delta cells. *NGN3* and *MAFB*, which are known to promote endocrine cell differentiation, were predicted targets of miR-148a-3p in alpha, beta, and PP cells, supporting a potential role for this miRNA in the differentiation of exocrine and endocrine cells.

### 2.5. Potential Roles of miR-7-5p, miR-148-3p, and miR-375 in the Human Pancreas According to Their Experimentally Validated Targets

Next, we looked at target genes of miR-7-5p, miR-148-3p, and miR-375 that are highly expressed in islets (islet-genes 5 RPKM and up, based on an unpublished RNA sequencing dataset of human islets, the list of islet-genes is provided in Appendix A) and the aforementioned list of essential genes for the pancreas (regardless of their expression level). We used multiMiR to retrieve the interactions between these miRNAs and their targets from validated targets documented in the integrated multiple microRNA-target databases [43] that contain experimentally validated miRNA-target interactions curated by miRecords [44], miRTarBase [45], and TarBase [46]. There are 920, 506, and 368 validated targets for miR-7-5p, miR-148-3p, and miR-375, respectively, in the list of islet-genes (Figure 5A). In agreement with the results for predicted targets, miR-375 still has the least number of validated targets, but in contrast to only a few common targets of the three miRNAs, there are 111 islet-genes and two essential genes (IRS1 and SOX4) that are targeted by all three miRNAs (SOX4 is also in the list of islet-genes) in the validated targets. Among the targeted essential genes, PSIP1 is targeted by both miR-375 and miR-148a-3p, GATA6 is targeted by both miR-7-5p and miR-148a-3p, ERRFI1 is targeted by both miR-375 and miR-7-5p, and all three genes are also in the list of islet-genes. miR-375 has 9 validated targets that are essential genes and 6 of them are in the list of islet-genes; miR-7-5p has 10 validated targets that are essential genes and 9 of them are in the list of islet-genes; miR-148a-3p has 9 validated targets that are essential genes and 4 of them are in the list of islet-genes (Figure 5B,C). Therefore, the three miRNAs have an almost equal number of validated targets in the list of essential genes, and some essential genes are targeted by two or all three miRNAs. Next, using their validated targets in the list of islet-genes, we performed a comparison KEGG pathway analysis. It is not surprising to find that miR-7-5p regulates more pathways than miR-148a-3p, and that both miR-7-5p and miR-148a-3p regulate more pathways than miR-375 does, probably because the former two have more validated targets in the list of islet-genes. This analysis also revealed that miR-7-5p regulates the insulin signaling pathway more significantly than miR-148a-3p, and miR-375 has less regulation for this pathway, indicating that miR-7-5p plays a more important role in insulin regulation than miR-148a-3p, and miR-375 seems to participate less in the insulin signaling pathway (Appendix A). However, all three miRNAs regulate the FoxO signaling pathway that regulates apoptosis, cell-cycle control, glucose metabolism, the AEG-RAGE signaling pathway in diabetic complications, and the EGFR signaling pathway (Figure 5D and Figure 6). The KEGG pathway analysis also revealed that the validated targets of the three miRNAs are most involved in glucose uptake, glycogenesis, proliferation, and differentiation—especially proliferation and differentiation, as revealed by Pathview (2017) analysis—and support their important regulatory roles in pancreas development [47] (Figure 6 and Appendix A).

Taken together, the three major pancreatic miRNAs may work together to regulate pancreatic cell development and cell differentiation by targeting different pancreatic genes and essential pancreatic genes.

## 3. Discussion

Taking advantage of our unique opportunity to access high-quality human pancreas tissue and our bias reduction small RNA profiling methodology, we have profiled small RNAs from eight paired human samples of acinar cells and islets and four rodent cell lines. Our small RNA profiling data of human pancreatic tissues showed that miR-148a-3p was highly expressed in acini and islets, being the most abundant miRNA in acini, and miR-148a-3p, miR-375, and miR-7-5p were found to be the three most abundant human islets miRNAs. Our data also showed that miR-7-5p is the most abundant miRNA in human islets, a finding that is distinct from published miRNA sequencing data and microarray data [16]. Furthermore, in contrast to the very low expression level of miR-148a-3p in both datasets of GSE52314 and GSE47720, our profiles showed that miR-148a-3p is the most abundant pancreatic miRNA and present in near-equal abundance to miR-375 and miR-7-5p in human islets. Multiple factors that are essential for pancreatic cell differentiation or pancreatic development have been predicted and/or validated to be the targets of these miRNAs, supporting the idea that combined targeting by miR-148a-3p, miR-375, and miR-7-5p could play a role in pancreatic cell development and differentiation.

The function of miRNAs closely correlates with their abundance in cells. Highly expressed miRNAs have more chances to find and bind their targets and higher potency to repress their targets [48]. The overall miRNA expression level is controlled by transcription, processing to mature forms, and miRNA half-life. miRNAs can be produced rapidly and persist for minutes to weeks [49]. However, due to the numerous types, small size, and variation in half-life, miRNA profiling is challenging using traditional cloning, RT-PCR, microarray, and deep sequencing technologies. Among them, cloning methodologies are labor-intensive, costly, and restricted to the identification of highly expressed miRNAs in the early era of miRNA studies. Both RT-PCR and microarray technologies have similar limitations in that they rely on known small RNA sequences to identify their presence in new samples. High-throughput sequencing technologies can profile both known sequences and novel sequences, simplify small RNA profiling, increase capacity, reveal different isoforms and their abundance, and reduce costs, and have become the preferred small RNA profiling methodology. However, potential bias in the sampling of small RNA sequences has resulted in inconsistent smRNAseq data.

Variations in small RNA profiling data from tissue samples are unavoidable due to the multiple cell types that make up complex tissues and organs. Variation in sample handling and RNA isolation methods also contributes to data heterogeneity. Our smRNAseq data showed that miR-7-5p, which easily degrades with a short half-life [50], was the most abundant miRNA in human islets, which has been shown to be a low abundance pancreatic miRNA in some published datasets, implying that the RNA samples we used for sequencing were of high quality and our sequencing protocol performed well in surveying all members of small RNAs. The high expression level of miR-148a-3p in the pancreas, elevated levels of miR-7-5p, and decreased levels of miR-148a-3p in endocrine cells compared to exocrine cells are compatible with the distinct proliferative rates of exocrine and endocrine cells. The meaning of the high levels of miR-148a-3p in acini or pancreas remains to be experimentally determined. Presumably, it may play a critical role in the pancreas since the majority of the pancreatic mass consists of acini and ductal cells [51].

In summary, reanalysis of published datasets and de nova analysis of human pancreata and rodent pancreas cell lines revealed discrepancies in miRNA expression patterns. The high abundance of miR-375, miR-148a-3p, and miR-7-5p in the pancreas revealed by our less biased small RNA profiling methodology using high-quality human samples, and the special targeting mechanism of miR-375 and its ability to bind Toll-like receptors [52], suggests possible regulatory roles by the three miRNAs in pancreatic development, cell differentiation, and diabetes. There is a limitation in that we did not profile small RNA from all types of pancreatic cells, such as ductal cells and all members of islets (alpha, beta, delta, epsilon, and PP cells) in the current study due to the heterogeneity in pancreatic cells caused by subgroups in each cell type [53] and the technical challenges in single cell smRNAseq analysis [54,55]. There is another limitation in our study in that the acini and islets we used were of high quality but not as pure as sorted cells. Therefore, cross contamination of acini by islets, or islets by acini were unavoidable. However, fresh tissues have the advantage of reduced RNA degradation over decomposed tissues. In this regard, single cell smRNAseq analysis, which also needs islets to be dissociated into single cells, will also have the risk of RNA degradation. Nonetheless, either acini or islet samples must go through the process of collagenase digestion of the pancreata. Therefore, how we processed the samples may have impacted the profiling data we have obtained in the current study. Regarding the most highly expressed miRNAs in acini or islets, the high level of miR-375 in acini may have some carryover from islets and the high level of miR-148a-3p in islets may have some carryover from acini. However, the almost 30-fold expression level of miR-7-5p (20.18%) in islets over in acini (0.68%) indicates that the cross contamination between acini and islets in our samples is very low. Finally, we only analyzed eight pairs of samples with high islets quality, enough for statistical analysis, but we have not been able to consider other factors that could influence the outcome of the profiling, including gender, age, race, BMI, etc., and the outcome may also be dependent on the protocol used for tissue preparation.

Based on the profiling data, we propose a model in which miRNA may control human pancreas development. Specifically, miR-375 may be specific for pancreas development, the expression of miR-148a-3p in the pancreas may guide lineage separation of exocrine cells from endocrine cells, and the expression of miR-7-5p may reenforce the fate of endocrine cells., Both miR-148a-3p and miR-7-5p or miR-7-5p alone may work with miR-375 to promote subtype cell differentiation in islets.

## 4. Materials and Methods

All methods were carried out in accordance with relevant guidelines and regulations.

### 4.1. RNA Isolation

Aliquots of islets and acini were frozen and stored at −80 °C for later RNA isolation. RNA was isolated from tissues and cultured cell lines using TRIzol (ThermoFisher, Waltham, MA, USA). The manufacturer’s instructions were followed throughout. RNA quality was determined, and amounts quantified using a Nanodrop and an Agilent Bioanalyzer.

### 4.2. Small RNA Deep Sequencing, Reads Processing, and Data Analysis

Small RNA deep sequencing was carried out using a customized protocol [25,26]. Briefly, 100 ng to 1.0 µg of RNA was used to construct small RNA libraries for single reads, flow cell cluster generation, and 51 cycle (51-nt) sequencing on a HiSeq 2500 (Illumina, San Diego, CA, USA). The sequence depth was 30M reads per sample, 8 samples per lane (by barcoding), and was performed by the Integrative Genomics Core of City of Hope National Medical Center. To construct small RNA libraries, the 5′ adaptor used in the Illumina (San Diego, CA, USA) Truseq small RNA deep sequencing protocol was replaced with a customized 5′ adaptor by adding 3 random nucleotides (nts) at the 3′ end of the original 5′ adaptor to reduce bias [25,26]. The 3′ adaptor provided in the kit was used for 3′ end ligation and sample barcoding.

Raw smRNAseq reads for each sample were separated by barcodes. Then the 5′ adaptor with the three random nts was removed prior to further processing and analyzing using miRge [12,13]. Unnormalized miRNA read counts generated by miRge were inputted into the Bioconductor package DESeq2 (version 1.30.0) in R (version 4.0.2) to generate the differential expressed miRNA results (islet versus acinus) at the default setting [the Wald-test was applied to assess the *p* value for differential gene expressions and the adjusted *p* value (*p*-adj) was obtained using the method of Benjamini and Hochberg] [56].

### 4.3. Small RNA qRT-PCR

We followed a published S-Poly(T) small RNA qRT-PCR detection protocol with some modification for multiplex in a reverse transcription step [26,57]. Briefly, 100 ng DNase I treated total RNA was poly-A tailed using the poly-A tailing kit from Epicentre (Madison, WI, USA). Since miR-26a-3p and miR-27a-3p had similar abundance in acini and islets, miRNA-26a-3p was selected as an RNA sample loading control to normalize relative expression of other miRNA, and miR-27a-3p was used to validate similar miRNA-26a-3p expression levels across different samples. Reverse transcription primers for all eight miRNAs were pooled and used in one reverse transcription reaction. Reverse transcription primers for miRNAs were designed according to the most abundant isoforms of a given miRNA in miRBase. All oligoes were from Integrated DNA Technologies (Coralville, IA, USA). Universal TaqMan probe and PCR mixtures were employed as published [57]. The ΔCq (quantitation cycle) value of an miNRA relative to miR-26a-3p in each sample was calculated as Cq_miRNA_-Cq_miRNA-26a-3p_. The value of 2^ΔCt^ was calculated as the relative fold change of an miRNA to miRNA-26a-3p. miRNA qRT-PCR primers are list in Appendix A.

### 4.4. Isolation of Human Pancreatic Islets and Acinar Cells

Brain-dead cadaveric donor pancreata were obtained from a local organ procurement organization. Organs were shipped after flushing with cold University of Wisconsin preservation solution. Pancreatic islet and acinar isolation protocol was followed to obtain these cells as described [58]. Briefly, pancreata were digested using collagenase supplemented with either thermolysin or neutral protease. Then, the digested pancreatic tissue was collected, washed, and purified in a cooled COBE 2991 cell processor (COBE Laboratories Inc., Lakewood, CA, USA). After islet purification, the acinar cells were collected from the remaining digested pancreatic tissue. As a standard procedure, following each isolation, immunofluorescent staining was used to check the purity of human islets and acinar cells. Islets were stained with insulin and acinar cells were stained with amylase to show the distinct cell population harvested during islets isolation (Appendix A). Acinar samples had <1% islets. Islets were cultured at 37 °C with 5% CO_2_ for 24–72 h before assessing purity. The use of human pancreatic tissues was approved by the Institutional Review Board of City of Hope National Medical Center. Informed consent was obtained from the legal next of kin of each donor. Donor information is provided in Table 1.

### 4.5. Cell Lines and Culture Conditions

Murine alpha-TC1 (ATCC #CRL-2934) and beta-TC-6 (ATCC # CRL-11506) cells were purchased from the American Type Culture Collection (Manassas, VA, USA). MIN6 cells were a kind gift from Dr. Jun-ichi Miyazaki (Osaka University Medical School, Osaka, Japan). Rat insulinoma INS-1 cells were a kind gift from Dr. Fouad Kandeel (City of Hope National Medical Center, Duarte, CA, USA). Both MIN-6 and INS-1 cell lines were not characterized for their identity.

All cell lines were cultured at 37 °C in a humidified atmosphere containing 5% CO_2_ in medium supplemented with 2 mM penicillin/streptomycin. Alpha-TC1 cells were cultured in DMEM supplemented with 10% heat inactivated fetal bovine serum (FBS), 15 mM HEPES, 0.1 mM non-essential amino acids, 0.02% BSA, 0.5 g/L Sodium Bicarbonate, and 2.0 g/L Glucose. Beta-TC-6 and MIN6 cells were cultured in DMEM supplemented with 15% heat inactivated FBS. INS-1 cells were cultured in RPMI-1640 supplemented with 10% FBS, 10 mM Hepes, 2 mM L-glutamine, 2 g/L Glucose, 1 mM sodium pyruvate, and 0.05 mM 2-mercaptoethanol.

### 4.6. Computational Resources

R version 4.02, R studio, and many R or Bioconductor software packages were used for data analysis and visualization. ComplexHeatmap was used to generate the heatmaps [59], the EnhancedVolcano R package was used to generate the volcano plots [60], and clusterProfiler for KEGG pathway analysis [61]. Pathview was used to generate KEGG graphs [47]. Other smRNAseq analysis tools and miRNA targets prediction or document tools used included miRmine [15], miRDB [38], miRwalk [39], TargetScan [35], and multiMiR [43].

### 4.7. Statistical Analyses

Inferential statistics of differential miRNA expression in islets versus acinar cells were calculated using the default setting in DESeq2. Descriptive statistics in miRNA qPCR data or comparison of one miRNA in islets versus acini or in two different data sets were calculated by two tailed *t*-tests. Unless specified in the figure legends, *p*-adj < 0.05 for DESeq2 data or *p*-value < 0.05 for others was used as the cutoff for significance in gene expression. Error is expressed as SD.

## Figures and Tables

**Figure 2 ncrna-09-00020-f002:**
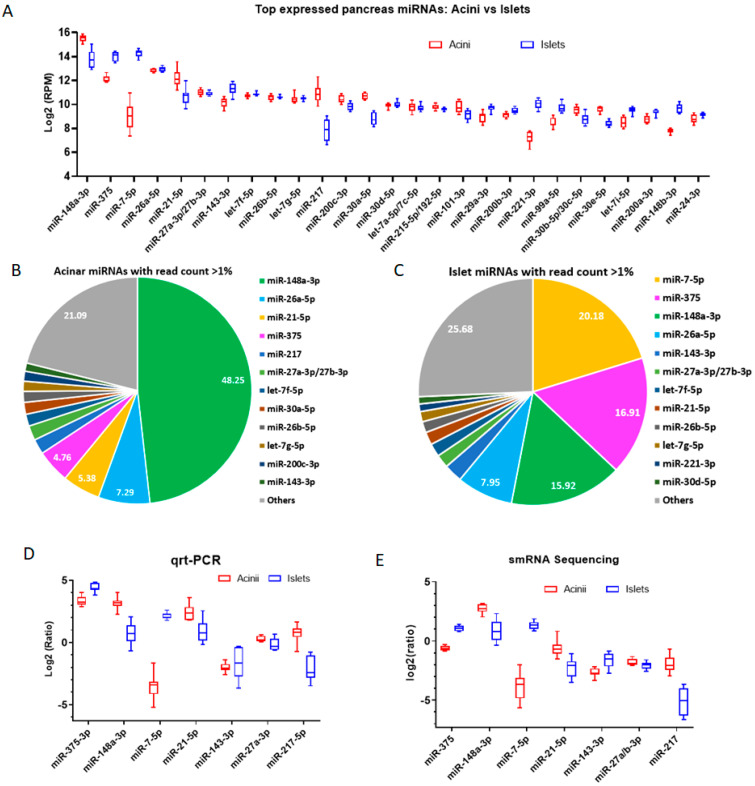
miRNA expression in acini versus islets. (**A**) Bar plot of top 25 highly expressed miRNAs in acini and islets cells measured by log_2_ value of read per million reads (RPM). (**B**) Pie chart plot of miRNAs in acinar cells having an average read count over 1% of total miRNA read counts. (**C**) Pie chart plot of miRNAs in islets having an average read count over 1% of total miRNA read counts. (**D**) Relative smRNAseq expression in acini and islets by qRT-PCR—plot of log_2_ ratio of qRT-PCR values of seven miRNAs to miR-26a-3p. (**E**) Relative smRNAseq expression in acini and islets by smRNAseq—plot of log_2_ ratio of RPM values of seven miRNAs to miR-26a-3p. In panels (**A**,**D**,**E**): Y-axis values represent the mean value of all eight human samples and the error bar represents standard derivation. In panel (**B**): a bar chart value represents the mean percentage value of all eight human samples in each tissue.

**Figure 3 ncrna-09-00020-f003:**
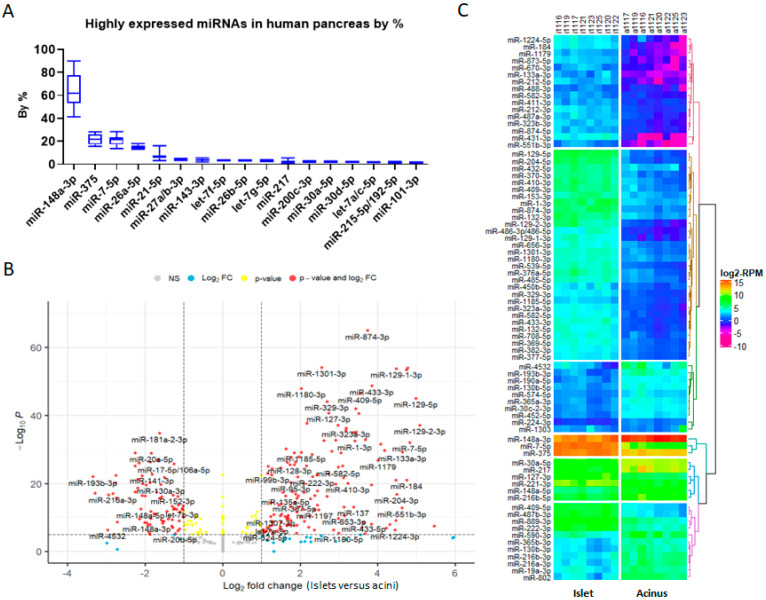
Pancreas miRNAs and differentially expressed miRNAs in islets versus acini. (**A**) Bar plot of top 14 highly expressed miRNAs in the human pancreas. (**B**) Volcano plot of differentially expressed miRNAs (islet versus acinus). (**C**) Heatmap plot of miRNAs with abs(log_2_FC) ≥ 2, padj ≤ 0.05, and DEseq2 calculated RPM (baseMean) ≥ 100. miR-148-3p and miR-375, which have abs(log_2_FC) < 2, were added for comparison.

**Figure 4 ncrna-09-00020-f004:**
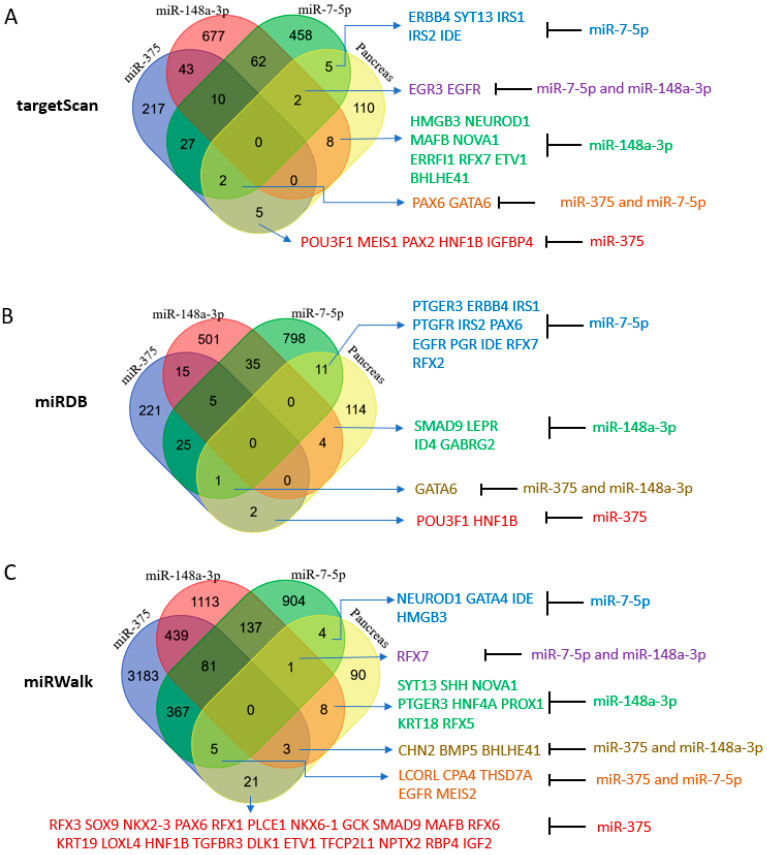
Pancreas-essential genes are targets of miR-148a-3p, miR-375, and miR-7-5p. (**A**) Venn Diagram analysis of TargetScan predicted miRNA targets that are pancreas-essential genes. (**B**) Venn Diagram analysis of miRDB predicted miRNA targets that are pancreas-essential genes. (**C**) Venn Diagram analysis of miRWalk predicted miRNA targets that are pancreas-essential genes.

**Figure 5 ncrna-09-00020-f005:**
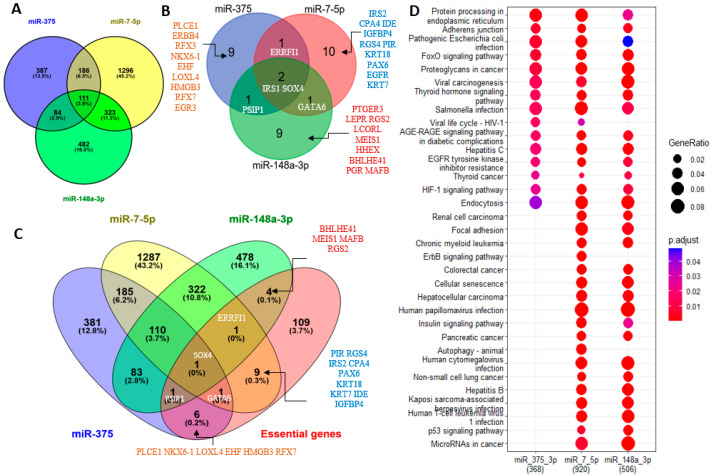
Islet-genes and pancreas-essential genes are experimentally validated targets of miR-148a-3p, miR-375, and miR-7-5p. (**A**) Venn Diagram analysis of islet-genes that are experimentally validated targets. (**B**) Venn Diagram analysis of pancreas-essential genes that are experimentally validated targets. (**C**) Venn Diagram analysis of islet-genes and pancreas-essential genes that are experimentally validated targets. (**D**). Comparison KEGG pathway analysis of islet-genes that are validated targets of miR-148a-3p, miR-375, and miR-7-5p.

**Figure 6 ncrna-09-00020-f006:**
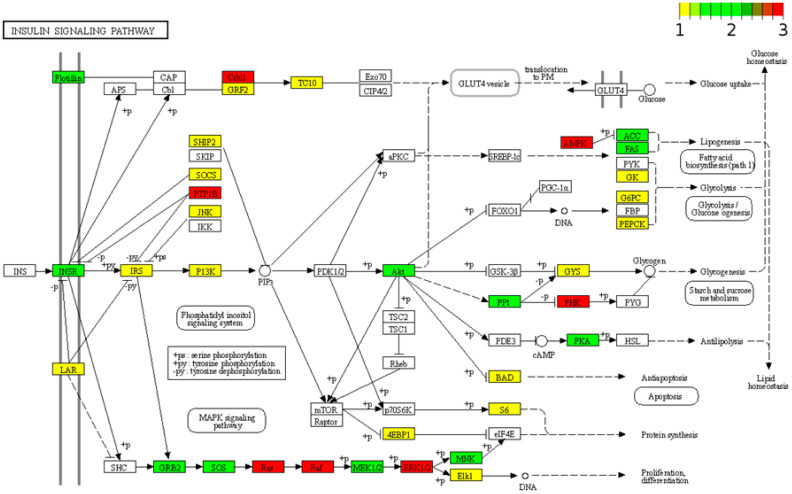
KEGG pathway view of islet-genes that are experimentally validated targets of miR-148a-3p, miR-375, and miR-7-5p. All islet-genes that are experimentally validated targets of miR-148a-3p, miR-375, and miR-7 were put into Pathview to generate a KEGG pathway graph of the insulin signaling pathway (hsa04910). If the node of a protein is labeled in white, no targets from the three miRNAs is involved in regulating this protein. Nodes labeled from yellow to green to red indicate that an increased number of targets are correlated with the protein at this node in the pathway.

**Table 1 ncrna-09-00020-t001:** Pancreas tissue donors’ information.

Isolation Number	Age (Years)	Race	Sex	HbA1c (%)	BMI	Cause of Death	UNOS#	OPO
Hu 1116	52	Caucasian	M	5	20.7	HT	AFEJ152	OneLegacy
Hu 1117	46	Hispanic	F	5.6	29.4	HT	AFEQ297	OneLegacy
Hu 1119	23	Caucasian	M	6.1	26.6	HT	AFEU424	OneLegacy
Hu 1120	47	Asian	F	5.5	22.4	CVA	AFE3481	OneLegacy
Hu 1121	25	Hispanic	M	5.6	32	HT	AFFE289	OneLegacy
Hu 1122	29	Hispanic	M	5	25	HT	AFFG219	OneLegacy
Hu 1123	51	Hispanic	M	5.4	35.6	CVA	AFFK170	OneLegacy
Hu1125	54	Hispanic	M	5.8	23.6	CVA	AFFW462	OneLegacy

HT: Head trauma, CVA: Cerebrovascular accident, OPO: Organ Procurement Organization, UNOS#: Universal donor IDs.

## Data Availability

All raw and some processed sequencing data generated in this study was submitted to the NCBI Gene Expression Omnibus (GEO; https://www.ncbi.nlm.nih.gov/geo/) under accession number GSE181066.

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
