# Peer review of "Reassessing the Abundance of miRNAs in the Human Pancreas and Rodent Cell Lines and Its Implication"

_ncrna, 2023, doi:10.3390/ncrna9020020_

Round 1

Reviewer 1 Report

Dear Authors,

first of all congratulations for your interesting work and effort. I hope that my comments will further help you to improve the paper, which already is very good. 

Lines 29-35 --> it would be better to have a picture depicting this sequence of events, additionally

Figure 1 --> abbreviations used in this table must be explained

Lines 203-209 --> How do we know that? Have you analysed many samples from many patients, or using single-cell technology? Not clear. 

Figure 2A --> Y axis numbers nor clear, it's not recommended to use E values on a graph. 

Figure 2D --> Needs reorganisation, too chaotic in a current form

Figure 3B --> Needs reorganisation, too chaotic in a current form

Discussion --> expanding welcome, especially in terms of clinical significance, potential drug targeting (maybe there are already some clinical trials?)

Reviewer 2 Report

The manuscript of Sun et al demonstrates microRNAs profile in Pancreatic cells, subdivided into acinus and islet, together with rodent Pancreatic cell lines. The manuscript is fascinating, for sure is important to rise variations and bias in microRNAs profile for Pancreatic cells between Scientific works, however, I can not see the novelty in data or technology. More than that, the human Pancreatic cells used in this manuscript were from adult humans, so the Pancreatic cells were physiologically balanced, demonstrating microRNAs that are presented in this context. In a conclusion, the authors stated “the three major pancreatic miRNAs may work together to regulate pancreatic cell development and cell differentiation by targeting different pancreatic genes and essential pancreatic genes.”, in fact, they do not demonstrate this and do not work with cell development and differentiation. In conclusion, I considered that the data demonstrated are not suitable for this journal and have some other considerations listed above:

Do the authors have MIN6 and INS-1 cell authentication certificates?

I agree with the authors that studies using human clinical samples are quite complex and the conclusion should be very careful. For the manuscript aspect, miRNA profile from Pancreatic cells presents bias and variations between manuscripts as stated: “Among all confounding factors, smRNAseq methodologies were found to be the major cause of result bias in small RNA profiling.” However, clinical sample characteristics are such an important factor. The reference [16] refers to a microarray protocol using just 3 human samples, showing no sample characteristics like age, gender, ethnicity, and healthy condition. The reference [17] is looking for microRNAs profile in human pancreatic development, a different context of this manuscript.

Figure 2: Figure A was not clear to me about the Y axis. Is absolute quantification? And how about standard deviation? Because this represents a mean of 8 human samples?  Figures A and B+C represent the same result. Show just one on the manuscript.

The authors stated that miR-26a-5p was chosen for normalization due to its similar expression level. However several microRNAs had the same similar level, like miR-27a-3p, let-7f-5p and miR-26b-5p. Normalization is a complex process and the usage of several normalization miRs is a nice procedure, using software like Genorm.

Reviewer 3 Report

Guihua et al compare in their publication the abundance of different miRNA in the islets of Langerhans. First by comparing different published datasets and tools describing a discrepancy of highly abundant miRNAs in those datasets mainly independent from the use of different tools for analysis.

Therefore they aimed to define a new miRNA-Sequencing technic which is able to overcome the discrepancy compared to other datasets. This technique was than applied to fresh human islets and acinar cells. However, when comparing to qPCR-results the new developed sequencing technique still shows some inconsistent data. This is claimed to be reasoned by isomirs.

Afterwards a new hierarchy of highly expressed miRNAs for the Langerhans islets in humans and mice is constructed based on the new sequencing approach.

Finally, by use of different target prediction tools and databases including experimentally validated miRNA targets, the role of those targets of the most highly expressed miRNAs is analysed by Pathway enrichment analysis. Focussing on the insulin signalling pathway revealing, that a high number of targets are involved in the KEGG Insulin signalling pathway.

Overall this study is important and very interesting for a big field of scientist investigating not only non-conding RNAs but also pancreas development and diabetes. Giving a resource of reall highly abundant miRNAs in Langerhans islets. However, I’am missing a proof that is explaining the discrepancy between qPCR results and Sequencing. Only stating this might be due to isomirs is not enough to claim the developed sequencing strategy is better than methods before. From my point of view this is a critical point, necessary and should be proven or at least indicated via the analyse of isomirs in the sequencing data.

In generell I have the impression that the authors tend to overinterprate the data that was given by miRNA-target prediction. I would like to encourage them to use terms like “might”, “this indicates”, “this gives the impression” to clearify that this is not proofen.

Major points:

1.       The inconsistent data for miR-375 and miR-21-5p explained by isomirs has to be proofen, which should not be a problem, as you have the sequencing data available. Showing the expression levels of those isomirs for miR-375 and to see the saturation. Otherwise one of the central results of these analysis is hard to be considered as true.

2.       I think Figure4 is unessential for the here shown story and the supplemental might be a better fit for this additional information, as the focus is set clearly on the human data.

3.       miRNA target prediction could be much improved, as there is in generell a consens that miRNA target prediction includes a high number of FALSE positive results. Therefore often a overlap of different target prediction is used to identify true targets.

4.       It is crucial to understand from which databases and which kind of experiments the experimental validated targets come. The reason for this is that it is well known that miRNAs often have highly tissue specific interactions and for example high throughput methods like Hit-clips for liver not necessarily match to those of islets. I think this has to be stated more clearly to not overinterpret the data.

Minor points

1.       Review Table 1A/B The nomenclature that all values represent percentage is missing. It should be added at least in the color scale.

2.       For the comparison of qPCR with small RNA-sequencing results a correlation analysis between both tools would be a good fit to help arguing about the accuracy of the method.

3.       Overall the final statement is clearly overinterpreting the data to conclude the relevance for cell fate decision of single miRNAs much more is needed than only miRNA-sequencing of different populations and target prediction.

Round 2

Reviewer 2 Report

The manuscript gives a slightly different microRNAs profile in pancreatic cells, which can be direct players on pancreatic context and represent an improvement on basic science context.  

Reviewer 3 Report

I thank the authors to answer all my question in a satisfying manner :)